# Improved Wearable Devices for Dietary Assessment Using a New Camera System

**DOI:** 10.3390/s22208006

**Published:** 2022-10-20

**Authors:** Mingui Sun, Wenyan Jia, Guangzong Chen, Mingke Hou, Jiacheng Chen, Zhi-Hong Mao

**Affiliations:** 1Department of Neurological Surgery, University of Pittsburgh, Pittsburgh, PA 15260, USA; 2Department of Electrical & Computer Engineering, University of Pittsburgh, Pittsburgh, PA 15260, USA; 3Department of Bioengineering, University of Pittsburgh, Pittsburgh, PA 15260, USA; 4Department of Mechanical Engineering, University of Pittsburgh, Pittsburgh, PA 15260, USA

**Keywords:** wearable devices, hardware design, dietary assessment, circular images, camera orientation, artificial intelligence

## Abstract

An unhealthy diet is strongly linked to obesity and numerous chronic diseases. Currently, over two-thirds of American adults are overweight or obese. Although dietary assessment helps people improve nutrition and lifestyle, traditional methods for dietary assessment depend on self-report, which is inaccurate and often biased. In recent years, as electronics, information, and artificial intelligence (AI) technologies advanced rapidly, image-based objective dietary assessment using wearable electronic devices has become a powerful approach. However, research in this field has been focused on the developments of advanced algorithms to process image data. Few reports exist on the study of device hardware for the particular purpose of dietary assessment. In this work, we demonstrate that, with the current hardware design, there is a considerable risk of missing important dietary data owing to the common use of rectangular image screen and fixed camera orientation. We then present two designs of a new camera system to reduce data loss by generating circular images using rectangular image sensor chips. We also present a mechanical design that allows the camera orientation to be adjusted, adapting to differences among device wearers, such as gender, body height, and so on. Finally, we discuss the pros and cons of rectangular versus circular images with respect to information preservation and data processing using AI algorithms.

## 1. Introduction

Food is essential to support human life; conversely, an unhealthy diet is strongly linked to risks of chronic diseases, such as cardiovascular diseases, diabetes, and certain types of cancer [1]. The Global Burden of Disease Study has found that, among the top 17 risk factors, poor diet is overwhelmingly the top risk factor for human diseases [2]. To study the quality (healthiness) and quantity (energy intake) in people’s diet, scientists need tools to obtain accurate information about the foods/beverages consumed by an individual over a certain period of time (e.g., one week) along with the volume or weight (called “portion size” in dietetics) of each food. This type of evaluation is called a dietary assessment (DA). Currently, self-report is the most commonly used DA method [3,4,5]. In this method, the person being evaluated takes detailed notes for each food/beverage as soon as the food is consumed (for simplicity, from now on, we will not particularly mention “beverage” and consider it is a particular type of “food”). This method is called a food diary [3,4,5,6]. In another self-report method called 24-h recall [3,4,5,7], the person being evaluated recalls each food consumed during the past 24 h. This recall is traditionally performed in a person-to-person interview by a dietitian. In recent years, the use of a web- or app-based electronic platform is gaining popularity [8,9,10,11,12]. In both cases, a food database (e.g., the FNDDS database developed by USDA [13]) is used to obtain the amounts of nutrients and energy for each food. Although widely utilized, self-report depends on the memory and willingness of the person to provide accurate and complete food-intake information. However, numerous studies have found that people tend to over-report healthy foods, but under-report unhealthy foods [7,14,15]. This type of reporting error is called subjective bias. In addition, both food diary and 24-h recall are complex and tedious procedures. Thus, their “participant burden” is high [7].

To solve the problems of self-report, image-based DA tools using wearable devices (we call them DA wearables) have emerged [16,17,18,19,20,21,22,23,24,25,26,27,28,29]. The DA wearables are equipped with a camera, as shown in Figure 1. The lens of the camera is oriented downward, aiming at the food on a dining table. The camera takes pictures automatically at a pre-programmed rate (e.g., 1–6 s between consecutive pictures). The images obtained are either stored within the DA wearable or transmitted wirelessly to a companion smartphone, where the data are stored or relayed to a remote server. Next, foods are identified, segmented, and their volumes are estimated, assisted by image processing algorithms. Finally, the food names and portion sizes are provided to a food database to obtain the nutrient/energy information. When compared with the self-report, the DA wearable approach reduces both subjective bias and participant burden because DA is conducted from images rather than the individual’s reports. However, the individual must be willing to image his/her foods and permit a dietitian to observe them. Some people may have a privacy concern on the images within which the background scene and people (e.g., family members) may be recorded unintentionally. Therefore, the image-based method has limitations. These limitations may be mitigated when AI algorithms, instead of humans, are used to process image data automatically (to be discussed further in the Discussion section).

Although commercial body-worn cameras, such as Narrative, Autographer, Vicon Revue, VIEVU, and FrontRow, are available, they are generally unsuitable to be used as DA wearables because these commercial devices are mostly designed for public security (e.g., police) or entertainment (e.g., lifelogging) purposes. As a result, their camera is forward-looking, which cannot capture the food below the camera effectively. These commercial products may also suffer from at least one of the following problems: bulky size, limited picture storage, short battery life, narrow field of view, and/or unsuitable picture-taking rate. Currently, DA studies usually use custom-made wearables. These devices are worn in different ways, such as the Automatic Ingestion Monitor (AIM) clipped on one side of eyeglasses [20,21,22], the Ear-Worn attached to a single ear [23,28], and the eButton pinned onto the chest [24,25,26,28].

Regardless of the types of wearables, the common goal is to capture a complete scene of foods on the table as a miss or an incomplete view of food results in a DA error. However, picture-taking by current DA wearables is self-activated (i.e., they do so by “random shoots”). The only way to minimize loss of food in images is to increase the coverage of the camera’s lens, i.e., to enlarge the field of view (FOV). In recent years, as mobile technologies advance, small and physically short camera modules (which makes the device thin) have become available [30,31]. Some of them have a FOV close to 180, which, theoretically, can capture the entire scene in front of the camera. However, there are two significant problems when these camera modules are used for DA wearables. First, these modules produce rectangular images at the output, which represents a crop of the circular FOV provided by the camera lens. A food may be outside the cropped region or cut by the cropping. Second, to obtain the best image quality and minimize content loss, the camera lens should be oriented to the direction at which foods are most likely to appear. However, this orientation depends on many factors, such as the wearer’s height, wearing location in the body, and/or the heights of the table and chair. Currently, these two problems have not yet been solved. All current DA wearables use imaging sensors with a rectangular screen and their camera is fix-mounted onto the device case without an adjustment mechanism.

In this work, we challenge the traditional camera design of DA wearables. A new camera system is presented to produce circular images instead of rectangular ones. We then present a new mechanical design that allows adjustment of camera orientation. Practical formulas are also provided to aid in the camera system design. Our new camera system produces more complete food intake information that increases DA accuracy.

## 2. Circular vs. Rectangular Images

In terms of picture-taking, there is a significant difference between a hand-held camera and a wearable one. While a hand-held camera is controlled manually (“aim and shoot”), the current DA wearable camera takes pictures without scene selection. When a rectangular frame is used by a DA wearable, it causes three major problems, as described below in detail.

### 2.1. Loss of Image Content

The loss of image content by a rectangular screen is illustrated in Figure 2. In both panels, the red circle represents the image field (IF) in the image plane, which is a plane within the camera where the sensor chip is placed. The IF is circular because the camera lens is always round. To produce a rectangular image from a round lens, the circular IF in the image plane must be cropped. For an image with a 4:3 screen ratio (left panel), this cropping wastes 38.9% of the available IF (i.e., the four white regions within the red circle are wasted). For a 16:9 image frame (right panel), the wasted area increases to 45.6%. Because, as mentioned earlier, the DA wearable “shoots” blindly, the risk of information loss due to the cropping effect is very high. To illustrate, let us imagine the case where one is blindfolded but left with a rectangular opening to observe the world. Certainly, it will be more likely to miss a target of interest compared with the case without such a blindfold.

### 2.2. Variable Field of View

The field of view (FOV) of a camera is defined as
(1)$$\mathrm{FOV}=2\tan^{-1}\frac{d}{2f}\;$$
where *d* and *f* represent the diagonal length of the image sensor chip and the focal length of the camera, respectively. For a rectangular image, the effective FOV in the horizontal or vertical direction is smaller than the diagonal direction. For example, for a camera with a 60° FOV and 4:3 screen ratio, the effective FOVs are only 49.6° (horizontal) and 38.2° (vertical).

### 2.3. Effect of Image Distortion

In the DA application, a wide-angle lens is highly desirable to obtain a large FOV. This type of lens bends the lights from the objects in the outlying regions of the scene (i.e., regions near the boundaries of the image) so that a flat-surface image sensor chip can record these lights. This process results in a barrel distortion, as shown in Figure 3 (left column), which needs to be corrected by a process called “undistortion”. Numerous undistortion methods have been reported [32,33,34,35]. Most methods utilize a distortion model (e.g., a stereo-graphic model). If the model is direction-invariant from the optical axis (usually located at the center of the image), the circular image after undistorting is still circular. Otherwise, the output image shape changes, but, in general, the change is not excessive (exemplified in Figure 3, top row). This shape-invariant or nearly invariant property is attractive for DA wearables because it implies that the image does not need post-processing after undistorting, preserving the information in the image. On the other hand, for rectangular images, the image after undistorting has a significant unnatural shape change (Figure 3, bottom row), which must be cropped, implying some information loss.

The circular image has another significant advantage for DA wearables. In practice, a DA wearable is often worn with an angle from its leveled position unintentionally. If this angle is large, the acquired images need to be rotated (re-levelled). If the input image is circular, the result after rotation is still circular without the need for cropping. In contrast, for rectangular images, the result after rotation must be cropped for a leveled presentation, which, again, leads to information loss.

## 3. Circular Image Generation

To produce circular images for DA wearables, an obvious method is to use a circular sensor chip (e.g., a circular CMOS chip) that has the same diameter as the circular IF. However, we have not found any manufacturers making circular image sensor chips. As a result, circular images must be produced from existing rectangular sensor chips. We have studied two methods to generate circular images. One is to rematch sensor chips and lens, and the other is to use an ultra-wide-angle fisheye lens only recently made available.

### 3.1. Rematch between Sensor Chip and Lens

The rematch method is illustrated in Figure 4. In the current design (left panel), the rectangular CCD or CMOS image sensor chip is placed within the circular IF produced by the round lens. In order to capture the entire image content within the circular IF, we rematch the chip and lens pair using a larger sensor chip, placed at the same distance to the optical center as the chip of the original size. The new chip can be determined according to Figure 5, where $d_{1},\;d_{2},h_{1},\;h_{2}$, and η represent the diagonal length of the original chip, the diagonal length of the rematched chip, the height of the original chip, the height of the rematched chip, and the screen ratio of both chips (for simplicity, here, we assume that the screen ratios of sensor chips before and after the rematch are unchanged), respectively. From Figure 5, to cover the entire circular field, the rematch must satisfy the following inequality:(2)$$d_{2}\;\geq \;\sqrt{h_{2}^{2}\;+\;{\left(\eta h_{2}\right)}^{2}}$$

As $d_{1}$ and $h_{2}$ are both diameters of the circle, we have $d_{1}=h_{2}$. Equation (2) then reduces to the following:(3)$$d_{2}\;\geq \;d_{1\;}\sqrt{1+\eta^{2}}$$

For example, let the original image sensor chip have a diagonal length $d_{1}$ of 1/7″ and the screen ratio η of both chips be 4:3. To obtain a circular image with a diameter of 1/7″ in the image plane, we must satisfy
(4)$$d_{2}\;\geq \;\left(\frac{1}{7}\right)\sqrt{1\;+\;{\left(\frac{4}{3}\right)}^{2}}\;=\;\frac{5}{21}$$

We may choose $d_{2}\;=\;\frac{5}{20}\;=\;\frac{1}{4}$, i.e., a 1/4″ sensor chip placed in the same image plane will allow the rematched imaging sensor to produce complete circular images with a diameter of 1/7″. 

We point out that the inequality in Equation (3) and the example provide a theoretical guideline only. In practice, the die size and the effective size of the sensor chip are often different. Therefore, in actual design, we recommend a thorough study of the datasheets of both the original and rematched chips. We also point out that the rematch method will waste some pixels (those in white regions in the right panel of Figure 4). In addition, storing circular images using the rectangular image format is less efficient because of the empty regions. To reduce the inefficiency, we suggest choosing a screen ratio η as close to 1 as possible (e.g., the 4:3 ratio is better than the 16:9 ratio). Further, as images are commonly stored in a compressed format (e.g., JPEG), the storage efficiency increases significantly if a constant pixel value (e.g., zero) is pre-written into the empty region before compression.

Another important question is how to choose the resolution of the circular image so that the details of the image are preserved. Let $\sigma_{1}$ and $\sigma_{2}$ represent the pixel densities (in pixels/mm) of the two image sensor chips before and after the rematch, respectively. Note that sensor chip manufacturers often provide the reciprocal of the pixel density, called “pixel size”, in the chip’s datasheet. Let $N_{1}$ and $N_{2}$ be the numbers of pixels of the two sensor chips before and after the rematch, respectively. From Figure 5, we have
(5)$$N_{1}\;=\;\eta h_{1}^{2}\sigma_{1}^{2}\;\mathrm{and}\;N_{2}\;=\;\eta h_{2}^{2}\sigma_{2}^{2}$$

For simplicity, let us consider only the largest circle contained in the outer rectangle, as shown in Figure 5, which corresponds to the equal sign in (3). As $d_{1}$ and $h_{2}$ are both diameters of the circle, we have
(6)$$h_{2}=d_{1}\;=\sqrt{\left(1+\eta^{2}\right)}\;h_{1}$$

Combining Equations (5) and (6)
(7)$$N_{2}{=}_{\;}\left(1+\eta^{2}\right)\frac{\sigma_{2}^{2}}{\sigma_{1}^{2}}N_{1}$$

Given $\sigma_{1}$ in the original image, let us discuss two scenarios to choose $\sigma_{2}$ in the rematched image. If one would like to keep the original image resolution unchanged, it requires $\sigma_{2}=\sigma_{1}$. Then, the number of pixels $N_{2}$ is $1+\eta^{2}$ times larger than that of $N_{1}$. For example, for the 4:3 screen ratio, $N_{2}$ is around 2.78 times larger than $N_{1}$. If this choice causes memory or data handling problems in the electronic hardware of the DA wearable, one may choose $\sigma_{2}$ by requiring equal number of pixels in the original image (the small rectangle in Figure 5) and the circular region in the rematched image (the circle in Figure 5). This is equivalent to
(8)$$N_{1}=\eta h_{1}^{2}\sigma_{1}^{2}=\pi {\left(\frac{h_{2}}{2}\right)}^{2}\sigma_{2}^{2}$$

Combining Equations (6) and (8), we can establish the relationship between $\sigma_{1}$ and $\sigma_{2}$
(9)$$\sigma_{2}=\sigma_{1}\sqrt{\frac{4\eta }{\pi (1+\eta^{2})}}$$

For the η = 4:3 screen ratio, we obtain $\sigma_{2}=0.78\sigma_{1}$. This result indicates a much smaller increase in data output (here, $N_{2}=1.69N_{1}$ vs. $N_{2}=2.78N_{1}$ in the previous case), but compromised by a 22% reduction in image resolution.

The sensor chip rematch method has two major advantages: (1) it allows a wide choice of FOV to satisfy the needs of different DA wearables and applications; and (2) for thinner DA wearables, the rematch method appears to be more suitable because the fisheye lens usually has a larger axial length that makes a DA wearable thicker, affecting its wearability. However, the rematch method has a disadvantage in that it may be difficult to combine a suitable pair of commercial lens and sensor chip with a correct lens mount thread size (e.g., M7). If this becomes a problem, a lens seat, including a polyimide ribbon connector (Figure 6), could be custom-made. Nevertheless, this approach is more expensive and could require longer design–test cycles.

### 3.2. Utilizing a Fisheye Lens

The second method to produce circular images is to use a commercial fisheye lens. Previously, a fisheye lens was usually long and heavy, unsuitable for use by DA wearables, which cannot be made cumbersome. In recent years, lens technology has improved significantly, and smaller and shorter fisheye lenses are now available. Despite the improvements, these lenses are still generally longer and heavier than non-fisheye lenses. The left panel in Figure 7a shows a fisheye lens that we have tested. This M7-lens (Type M7-1-08-Y, Nuoweian Inc., Shenzhen, China) has a focal length 1.08 mm, weight of 2.1 g, and height of 10.7 mm (about 15 mm after threading on a lens seat shown in Figure 6). Figure 7b,c shows two raw images obtained using this lens. It can be observed that these images are not completely circular. Portions of the image along the narrower direction of the image are missing. This phenomenon is quite common in small fisheye lenses. Although small portions of the circular field are lost on both sides, the empty regions outside the circular field are smaller than those shown in Figure 2, indicating a more efficient use of the active rectangular area of the sensor chip.

### 3.3. Comparisons between Circular and Rectangular Images 

To demonstrate the benefits of the new camera system, we compare, using real-world data, the results of circular images in the new system and rectangular images in the existing system. Figure 8a shows four consecutive FOVs (blue circles) calculated according to the polynomial radio distortion model [33,35,36,37], given by
(10)$$f\left(\rho \right)=a_{0}+a_{2}\rho^{2}+\ldots +a_{n}\rho^{n}$$
where f(ρ) is a mapping function determined by the particular lens construction (in our case, the Type M7-1-08-Y lens); *ρ* is the radial Euclidean distance from the image center in the sensor plane; and $a_{0},a_{2},\cdots ,a_{n}$ are the polynomial coefficients. According to this model, the points of the scene along the ray emanating from the optical center and passing through 3D point (u,v, f(ρ)) are mapped to point (u,v) on the imaging plane, with $\rho =\sqrt{u^{2}+v^{2}}\;.\;$ The polynomial coefficients are determined by a calibration process using a checkerboard phantom [33,35]. The FOV corresponding to each point (u,v) is calculated as follows:(11)$$FOV=2\tan^{-1}\frac{\rho }{f\left(\rho \right)}$$

The four blue circles in Figure 8a present circular image domains with FOVs equal to 60°, 100°, 140°, and 180°. For each circle, the inscribed red rectangle represents the traditional rectangular image domain (assuming the 4:3 screen ratio). Figure 8b shows four real-world food-containing images superimposed with the same image domains in Figure 8a. It can be observed that, for most of these real-world scenarios, both rectangular images and circular images with smaller FOVs tend to lose information because of missing or cutting portions of foods. Under the same FOV, the loss by the rectangular image is much more significant than the loss by the circular image. To facilitate observation of the losses, the circular and rectangular images corresponding to the top-left image in Figure 8b are shown in Figure 8c,d, respectively, for all four FOVs (labeled). It can be observed that circular images, especially those with larger FOVs, preserve image contents well. On the other hand, rectangular images are subject to higher risks of missing important food information.

## 4. Lens Orientation Adjustment

Although, in theory, a 180° circular lens on a DA wearable will not miss foods in front of a wearer, there is a practical problem in the varying effective resolution of the image content. For example, for a five-megapixel camera with a common 4:3 screen ratio, the circular image has approximately three megapixels. This number appears to be sufficiently high; however, it is true only in the central region of the circular image. Owing to the high distortion of the fisheye lens, as observed in Figure 7 and Figure 8, the effective resolution may become insufficient in the image outskirts away from the center. Increasing the number of pixels of the sensor chip is a clear choice to solve this problem, but it implies a higher cost; higher power consumption by the camera, central processor, and the data transfer circuitry; and likely a bulkier device, affecting its wearability. Another choice is to improve device orientation so that the lens is oriented appropriately. For DA wearables, the lens should be downward looking at a certain angle. As a result, the food items on the table should appear in, or close to, the central region of the image. In practice, however, the optimal orientation changes with several factors and varies in different individuals. For the eButton pinned onto the chest, the wearer’s age, gender, body height, heights of the table and chair, and the wearing location (towards one side or near the center of the chest) all affect the camera orientation. It is thus necessary to allow the wearer to adjust the lens orientation after the device is worn. Currently, none of the reported DA wearables have this adjustment mechanism. Thus, we independently designed a mechanical structure for the eButton to support the adjustment. This structure (shown in the right panel of Figure 9) includes two rotating axes, a camera module enclosure mounted with the axes, and two fixtures screw-mounted onto the body of the DA wearable. This mechanical structure has been 3D printed from digital blueprints made using the SOLIDWORKS mechanical design software (Dassault Systems, Paris, France), implemented in a recent version of the eButton (left panel in Figure 9) and utilized in a large-scale DA study in Africa [28].

## 5. Discussion

In recent years, there has been a steady trend of applying AI algorithms to image data acquired by DA wearables [38,39,40,41,42,43,44,45]. The AI approach not only reduces the data processing burden on researchers and dieticians, but also provides a solution to the privacy problem described in the Introduction section. However, as opposed to the lens orientation adjustment mechanism, which has been implemented and field-tested, we have not yet utilized circular images for real-world DA. The main reason was the difficulty encountered in data processing using AI algorithms. Currently, the existing convolutional neural networks (CNNs), which are the central components of AI algorithms for food image processing, are almost all designed for rectangular images [38,43,46,47,48,49]. Although isolated studies have been conducted for circular images, e.g., [50,51,52], they require training using the same type of images, which are not widely available. Research is ongoing in our laboratory to solve these problems.

We point out that our study on circular images has a biomimetic motivation. As far as our knowledge goes, none of the known eyes in the animal kingdom produce rectangular views. We believe that round or nearly round eyes allow better understanding of the environment. Additionally, the original radar, sonar, or oscilloscope screens were all round [53]. The rectangular screen became dominant in the past century for a number of valid reasons, including the convenient rectangular image shape in historical photography, the film strips that were made in rolls, the photographic equipment of manufacturers’ and photo viewers’ preferences, the ease of square-block based image processing procedures, and so on. Despite some drawbacks of circular images, for DA wearables and a number of other image-based wearables (e.g., those for the blind or vision impaired) where reliability in target finding is important, we believe that, at least for now, the advantages of circular images outweigh their drawbacks. However, in the future when AI, camera technology, and electronics are further developed, the next-generation miniature camera may recognize targets of interest and rotate its lens automatically before picture-taking. It is an interesting but unanswered question whether the future wearable devices will have round “biological eye(s)” or rectangular “robotic eye(s)”.

## 6. Conclusions

In this work, we targeted the missing data problem in image-based dietary assessment using wearable devices. We demonstrated that views of food items could be cropped out when rectangular images are produced. We presented two methods for generating circular images that preserve information. We also designed a mechanical structure to adjust camera lens orientation to obtain data of higher quality. Our approach may lead to significant improvements in using image-based wearable devices for dietary assessment and other applications. However, for this approach to be successful, there is a strong need to develop AI algorithms to extract information from circular images.

## Figures and Tables

**Figure 1 sensors-22-08006-f001:**
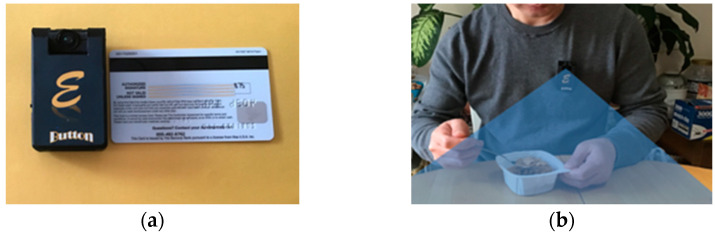
(**a**) The eButton is less than half the size of a credit card; (**b**) eButton takes pictures automatically during an eating event.

**Figure 2 sensors-22-08006-f002:**
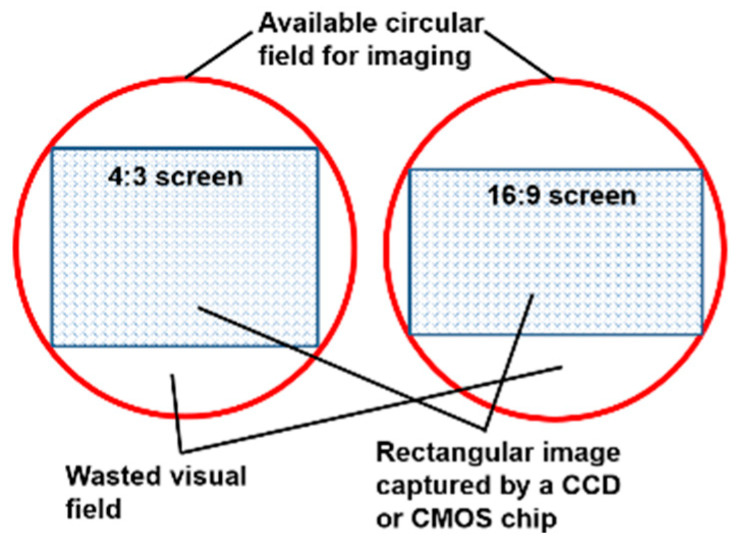
A rectangular image sensor chip is placed within a circular field in the image plane, capturing only part of the useable visual field.

**Figure 3 sensors-22-08006-f003:**
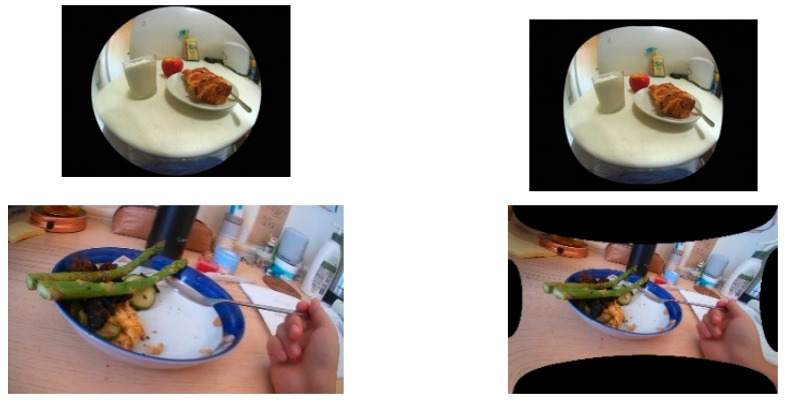
Inputs (**left**) and undistortion results (**right**). **Top row**: Circular image; **Bottom row**: Rectangular image.

**Figure 4 sensors-22-08006-f004:**
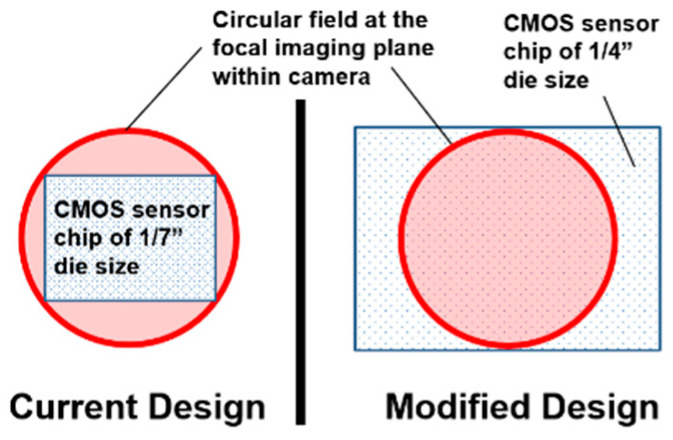
The current (**left**) and proposed (**right**) designs to acquire circular images using rectangular image sensor chips of different sizes.

**Figure 5 sensors-22-08006-f005:**
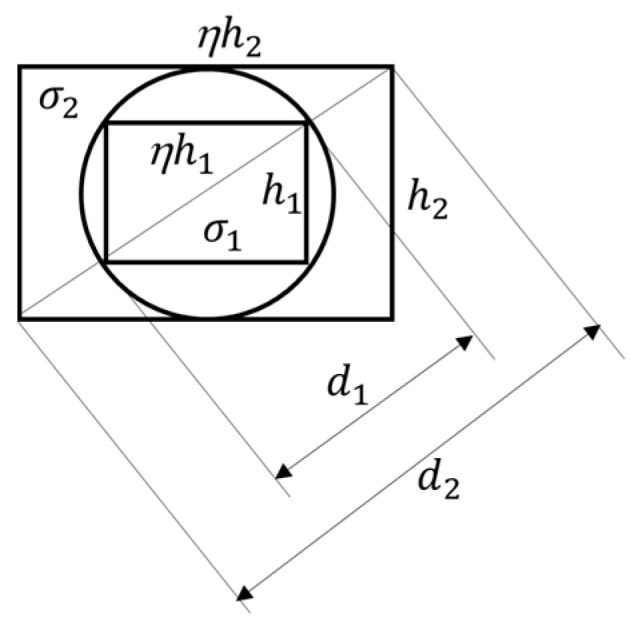
Geometric relationships in the rematch method.

**Figure 6 sensors-22-08006-f006:**
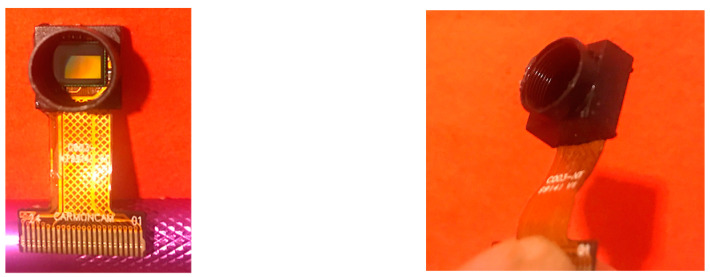
Lens seat with a ribbon connector. The image sensor can be seen from the left panel.

**Figure 7 sensors-22-08006-f007:**
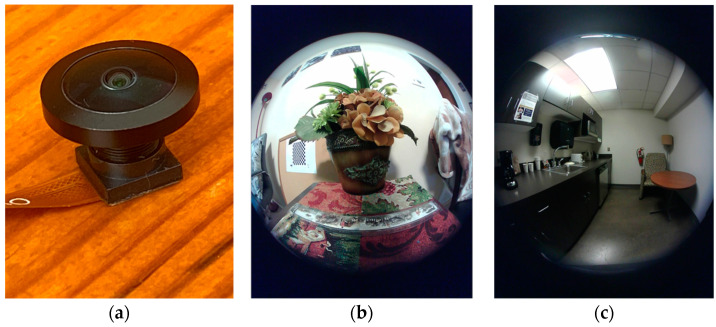
(**a**) Nuoweian fisheye lens, (**b**) and (**c**) raw images obtained by the lens.

**Figure 8 sensors-22-08006-f008:**
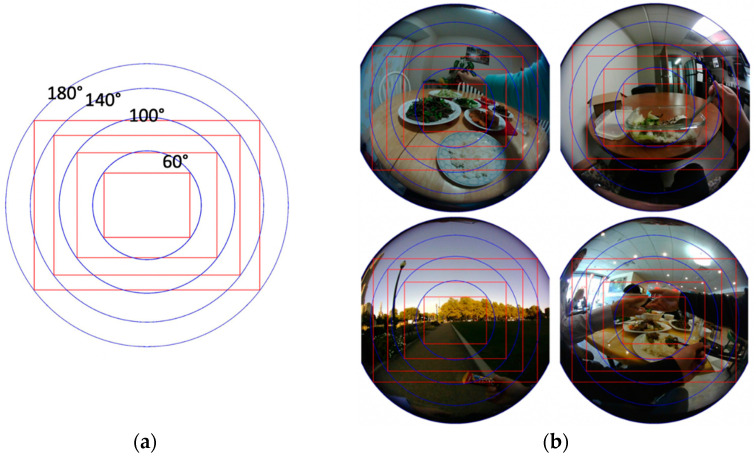
(**a**) Circular (blue) and rectangular (red, 4:3 screen ratio) image domains of different FOVs. (**b**) Real-world food-containing circular images acquired using our DA wearable (eButton) with a Type M7-1-08-Y fisheye lens. (**c**,**d**) Effects of circular and rectangular images corresponding to the FOVs in (**a**).

**Figure 9 sensors-22-08006-f009:**
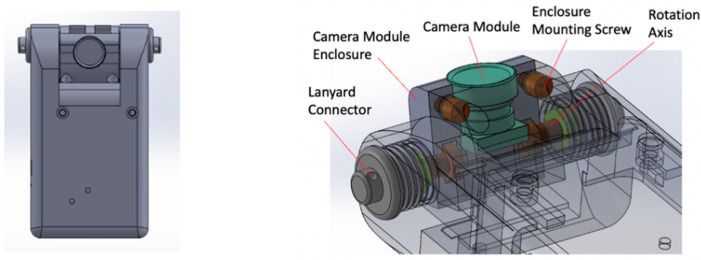
**Left panel:** Front view of the eButton with adjustable lens orientation; **Right panel:** Mechanical structure of the lens orientation adjustment assembly. For clarity, the device case is depicted in the transparent form.

## Data Availability

No new data were created or analyzed in this study. Data sharing is not applicable to this article.

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
