# Peer review of "Improved Wearable Devices for Dietary Assessment Using a New Camera System"

_sensors, 2022, doi:10.3390/s22208006_

Round 1

Reviewer 1 Report

This article presents two designs of the camera system, aiming to improve the current design in use of technological advances. The current manuscript requires to improve to a certain level as follows.

1.     Apart from the commercial devices, a briefly literature review of related wearable solutions for DA needs to be addressed. Evidences are needed to back up your sentence, such as “Over the years, there have been numerous reports on DA using wearable devices. 55 However, these reports have been focused on algorithm developments, few of them in- 56 investigating the hardware of DA wearables.”, “Global burden of disease study…” (lines 32-33); “Scientists do not have user-friendly tools, …” (line 34-36); “problem with existing products…”(lines 49-51), “Numerous reports on DA wearable devices, …” (lines 54-60, entire paragraph lines, please cite some of those numerous reports! Or at least a literature review of them); “algorithms being developed by our and other research groups…” (Please cite, lines 214-215); “isolated studies.. “ (lines 217-219); ”no known animal eyes produce rectangular vision (lines 221-223); “radar and sonar screens were round..” (lines 224-225)

2.     Why does DA wearable like eButton have potential to be beneficial to the public? What kinds of valuable dietary data are missed? What kind of negative impact might be derived from missing this kind of data?

3.     Why do the cropping wastes matter? If it is the main parameter to be addressed in this article, it is worthwhile raising this earlier, e.g., in the introduction section.

4.     Also, I like to see if you have evaluated the two designs. What have been improved and what can be done better?

5.     The conclusion needs to summary the findings, future improvement, practical implications, rather than simply addressing what the authors have done.

6.     Grammatical issue “Food is an important factor sustaining human life.”

Author Response

Thank you very much for your comments. 

Reviewer 2 Report

Esteemed Authors, 

It has been a great honor and a pleasantly challenging activity to review the article ”Improved Wearable Devices for Dietary Assessment Using a New Camera System.”

The authors' chosen topic is fascinating, given that the assessment of food consumption and nutritional and caloric intake are fundamental concerns of the scientific and medical world.

The paper is of high value due to its original character. It treats a specific subject of increased interest in medicine, nutrition, monitoring of food consumption, and finally, the dietary assessment using a new system developed by the authors.

The paper is well structured and possesses a high novelty character. The major components of the article – Introduction; Circular vs. Rectangular Images; Circular Image Generation; Lens Orientation Adjustment; Discussion and Conclusion - are organized judiciously and directly linked to one another.

The documentation is adequate but modest compared to scientific articles or other types of works consulted. For this reason, I suggest to the authors the development of this chapter by including in the list of bibliographic references other works representative of this field.

The goal of the conducted research is well specified and delineated. The working protocol is appropriate, and the analysis methods are correlated with the proposed objectives. All materials and methods are defined and described adequately.

The work also benefits from adequate iconographic support, materialized by eight figures. All eight figures are relevant and detail some technical aspects, a fact that greatly facilitates understanding the entire working mechanism of this assembly.

The bibliography is relevant for the article topic but presents some minor lacks regarding citations or mentions. To clarify some aspects, I would suggest that the authors write the references list evenly: for example, journal papers require either the complete journal name, the JCR abbreviation (in the case of ISI indexed or rated journals), or the ISO abbreviation (for BDI indexed journals); moreover, for journals, I suggest that the volume, number, and pages (as the case requires) be explicitly mentioned without exception. I would also recommend that more attention be paid when it comes to chapters from books and that the number of pages, the editor (publishing house), and other identification elements (link, etc.) be mentioned, regardless of the reference type.

Under these circumstances, the additional mention of the Digital Object Identifier (DOI) becomes optional.

The authors' presentation in the bibliographic reference list should be in alphabetical order: more attention is recommended regarding the authors' names, year of publication, and other identifying elements - I refer mainly to books, reports, etc.

The obtained results are interpreted correctly, and their practical value is visible.

The graphical representation of the results is adequate. As for the grammar of the paper, the text is very well written, with only two parts that would require minor corrections, as follows:

Page 3, line 104 – replace “after undistortion” with “after undistorting”;

Page 4, line 137 – replace “after rematch” with “after the rematch”.

Minor corrections and clarifications notwithstanding, the authors’ work and obtained results are highly commendable. They bring significant added value to the paper and may constitute a launching pad for further valuable studies.

The article can be accepted and published in the Sensors if the authors verify the paper and perform the required corrections.

Best Regards,

Reviewer

Author Response

Thank you very much for your comments. 

Reviewer 3 Report

This paper presents a new camera system aiming to improve dietary assessment. The authors clearly explain the motivation and need for such an approach and then in detail elaborate on the steps of the proposed methodology. The ideas presented in this article are valuable, however, the major drawback is the lack of experimental evaluation.

Major comments:

- No real dataset is collected to evaluate the effectiveness and value of the circular design of the captured images. Moreover, a theoretical analysis is needed to identify the minimum resolution of the captured images, so that the fragments of the circular images that would be lost when cropped to rectangular are valuable. Namely, you need to start with the minimum resolution of the cropped regions so that they contain enough detail to identify food or other relevant information, and then extrapolate this to the needed resolution of the camera sensors. This will present some lower/minimum requirements for the analysis to make sense.

Next, you need to capture some real-life with a prototype and illustrate the advantages of such a system.

Finally, a discussion from a social/psychological analysis of the device should be mentioned. Namely, suppose a person has a problem with obesity. In that case, he/she would already be likely to have a problem with self-esteem and be subject to judging or even bullying (especially if it's a teenager). Then adding one more gadget to monitor the food intake could be interpreted as "I have no self-control, don't know what/how much I'm eating, I need help from devices". I realize that this is harsh, and I do not support such views, but some people might not. Therefore, these aspects need to be carefully discussed, and addressed, if possible. At the very least, they need to be discussed in the limitation section, with some ideas on how to be mitigated.

Author Response

Thank you very much for your comments. 

Round 2

Reviewer 1 Report

The authors have answered my questions and modified the contents. No further suggestions.

Reviewer 3 Report

The authors have addressed all comments.